# Insoluble Network Skeleton and Soluble Components of Nylon 6,6-Sputtered Nanoparticles: Insights from Liquid-State and Solid-State NMR Analysis

**DOI:** 10.3390/nano14060497

**Published:** 2024-03-10

**Authors:** Julie Šťastná, Kateřina Škorvánková, Anna Kuzminova, Jan Hanuš, Lenka Hanyková, Ivan Krakovský, Pavel Solař

**Affiliations:** Department of Macromolecular Physics, Faculty of Mathematics and Physics, Charles University, V Holešovičkách 747/2, 180 00 Prague, Czech Republic; julie.stastna@matfyz.cuni.cz (J.Š.); katerina.skorvankova@gmail.com (K.Š.); anna.kuzminova@mff.cuni.cz (A.K.); jan.hanus@mff.cuni.cz (J.H.); hanykova@kmf.troja.mff.cuni.cz (L.H.); ivan.krakovsky@mff.cuni.cz (I.K.)

**Keywords:** gas aggregation source of nanoparticles, nanoparticles, plasma polymerisation, nuclear magnetic resonance, gel permeation chromatography, scanning electron microscopy

## Abstract

In this study, we performed a detailed analysis of -sputtered-nylon 6,6 plasma polymer nanoparticles (NPs). Following a previous study using standard techniques such as X-ray photoelectron spectroscopy (XPS) and Fourier transform infrared (FTIR) spectroscopy, we employed unconventional approaches, specifically solid- and liquid-state high-resolution nuclear magnetic resonance (NMR) spectroscopy, supplemented by gel permeation chromatography (GPC). Scanning electron microscopy (SEM) was also used to examine changes in the size of the NPs after contact with solvents and after heating. Our investigations revealed suspected strong binding and networking of the NPs, and a soluble monomer/oligomer phase was identified and characterised. This fraction is removable using solvent or heat treatment without significantly affecting the size of the NPs. Additionally, we suggested the chemical structure of this soluble phase. Our findings support the proposed rubber-like character of plasma polymer NPs and explain their strong tendency to reflect from substrates upon high-speed impact.

## 1. Introduction

Gas aggregation sources (GASs) are a popular method for the physical synthesis of nanoparticles (NPs). One of the most common types of magnetron-based GAS systems was introduced in the 1990s by Haberland et al. [1,2,3]. These sources have become popular because, due to the magnetron, they can produce a wide range of different types of metallic [4,5,6,7,8,9,10], metal oxide [11,12], polymer-like [13,14,15], or composite bi- and multi-component [16,17,18,19] NPs. 

Polymer NPs prepared using GASs were first reported by Wan et al. [13]. However, these NPs were prepared without plasma, using vacuum thermal decomposition. Using plasma is advantageous because it decreases the oily character of the deposits [20] and utilises a much wider range of precursors because plasma mediates polymerisation or repolymerisation of the source material [14,15,21,22]. 

Nylon is a promising precursor for fabricating amino-rich plasma polymers [23,24,25,26,27,28,29,30,31], which are desirable for many applications. For example, these polymers can selectively bind biomolecules and promote cell adhesion [23,32,33,34,35,36]. Aside from thin films produced at pressures around 1 Pa, it is also possible to prepare NPs by sputtering nylon in a GAS system at pressures of tens of pascals [14,15,37]. Plasma polymer NPs from sputtered nylon have been successfully deposited using two different systems—one with a planar magnetron [14] and the other with a semi-hollow magnetron [15].

This work used a semi-hollow magnetron, the set-up of which is described in Ref. [15], to deposit nylon-sputtered plasma NPs. In a subsequent study of these NPs [37], it was found that plasma polymer NPs readily reflect from the substrate if they impact with a high enough speed, suggesting the analogy of a rubber ball hitting a slightly sticky wall. While the reflection has been described, the mechanism could not be identified. Because plasma polymers are generally highly cross-linked, it may be expected that the plasma polymer NPs are reminiscent of rubber-like balls in the sense that each NP comprises a gelated polymer network with springy properties. To support this theory, the composition of the NPs must be determined.

Nylon-sputtered films and NPs are commonly investigated using X-ray photoelectron spectroscopy (XPS) and Fourier transform infrared (FTIR) spectroscopy [38,39]. These two methods were applied to the nylon NPs mentioned above. The results revealed only the elemental composition and the presence of chemical groups; they do not provide information about the overall structure. To investigate further, a GAS system, based on a semi-hollow magnetron with a nylon 6,6 target, as described in the abovementioned Refs. [15,37], was chosen for the experiments conducted in this paper using less conventional methods, that is, solid- and liquid-state high-resolution nuclear magnetic resonance (NMR) spectroscopy, complemented by gel permeation chromatography (GPC). 

To investigate the overall structure of the NPs, they were dissolved in various solvents and the soluble and insoluble parts were studied separately. This method has previously been successfully applied to the investigation of continuous thin films [40,41,42]. Liquid- and solid-state NMR spectroscopy have been infrequently used in this research field. However, employing NMR and GPC can provide interesting insights into the chemical structure of the samples [43]. In contrast to XPS, which has frequently been employed for analysing plasma polymer NPs, ^1^H NMR spectroscopy provides information about hydrogen atoms, enabling the differentiation of functional groups such as CH, CH_2_, and C=C. However, NMR spectroscopy has disadvantages, including the substantial amount of sample required and the restriction of analysis to the soluble portion when using liquid-state NMR. However, solid-state NMR can be used to analyse the entire sample, encompassing both the soluble and insoluble components [44]. Nevertheless, solid-state NMR spectroscopy primarily yields high-resolution carbon spectra and is highly sensitive to the mobility of the sample molecules [45]. 

This work aims to determine the composition of nylon-sputtered NPs deposited with the GAS system based on a semi-hollow magnetron used in our previous research, more precisely [15,37]. Knowledge of the overall structure is needed to better understand the problem of the reflection of the NPs from the substrate during deposition, as reported in Ref. [37]. Some of the results of this work may also be extended to other types of plasma polymer NPs and may offer ways to increase the efficiency of the production of plasma polymer NPs using gas aggregation sources.

## 2. Materials and Methods

The investigated NPs were deposited from a GAS based on a semi-hollow magnetron, in a set-up identical to the one described in Ref. [37] (Figure 1). The magnetron was prepared with a nylon 6,6 target (Goodfellow) with a 5 mm thick base and 1 mm thick cylinder walls. The magnetron was placed inside a standard ISO KF100 tube, capped with a conical lid and terminated with a tubular orifice (5.8 mm in diameter and 50 mm long), which together formed the aggregation chamber. The source was connected to a high-vacuum deposition chamber pumped using diffusion and rotary pumps. 

The magnetron was operated in pure argon (99.996% purity, Linde Gas, Prague, Czech Republic), supplied via a small hole in the centre of the magnetron using a flow controller (MF-1, range 20 sccm, MKS, Andover, MA, USA) at a pressure of 50 Pa and a flow rate of 19 sccm. The “sccm” stands for standard cubic centimetres per minute and indicates cubic centimetres of gas per minute, at a temperature of 273.15 K and a pressure of 10^5^ Pa. The discharge was operated using an RF generator (Dressler Cesar 600, Advanced Energy, Raunheim, Germany) connected to the powered electrode through a manually operated matchbox (MFJ-962D, MFJ Enterprises, Starkville, MS, USA). The pressure was measured in the aggregation and deposition chambers using a capacitance gauge (Baratron 626 C, range 133 Pa, MKS, Andover, MA, USA). 

The NPs were collected on glass slides positioned 3 cm from the orifice in the deposition chamber. To prevent NP reflection and to promote deposition, the NPs were decelerated [37]. The pressure in the deposition chamber was increased to 10 Pa by reducing the pumping rate using a butterfly valve between the deposition chamber and the diffusion pump. The deposits were subsequently wiped from the glass substrate into the solvents and sonicated for 10 min to disperse the NPs in the solvent. Then, the solution was left to rest for 24 h.

To visualise the NPs, they were deposited on silicon wafers at the orifice–substrate distance of 18 cm (for an appropriate deposition rate), at a deposition chamber pressure of 1.75 Pa. The NPs were then observed using scanning electron microscopy (SEM, JSM-7200F, JEOL, Tokio, Japan), operated in secondary electron mode.

To image the NPs in an SEM after they were dissolved in solvents, a drop of the solution was placed onto a silicon wafer, and the solvent was allowed to evaporate. To prevent excessive NP agglomeration during solvent evaporation, the following protocol was used: the silicon wafer was placed on a massive copper cylinder cooled with liquid nitrogen. A drop of the solution was placed on the wafer and then the whole apparatus was pumped down to 0.1 atm to accelerate the solvent evaporation. Although agglomeration could not be completely prevented, it was significantly reduced.

To determine the molar masses, the deposits were dissolved in tetrahydrofuran, the sediments were separated by decantation, and the soluble part was filtered using a Teflon filter with a 200 nm pore size. To perform GPC measurements, the solutions were adjusted to approximately 1% (*w*/*v*). The GPC measurements were performed using an Agilent 1260 Infinity GPC/SEC system equipped with a differential refractometric detector and a chromatographic column (7.5 mm × 300 mm, Varian Inc., Palo Alto, CA, USA) filled with 5 μm of sorbent particles of 100 Å pore size. Tetrahydrofuran (99.9%, HPLC grade, Sigma-Aldrich, St. Louis, MO, USA) was used as the mobile phase at a flow rate of 0.8 mL/min and a temperature of 30 °C. A universal calibration equation calculated from the GPC data obtained on polystyrene standards (Polymer Standards Service, Mainz, Germany) under the same conditions was used to determine the molar masses. 

In preparation for nuclear magnetic resonance (NMR) spectroscopy experiments, the NPs were separated from the glass substrate and dissolved in three different solvents (deuterated dimethyl sulfoxide (DMSO), deuterated acetic acid, and 5 wt% phenol in D_2_O at room temperature). In addition to the described 10 min of sonication and 24 h of rest, the mixture was sonicated for an additional 2 h before the measurements. Tetramethylsilane (TMS) was used as the internal standard; NMR shifts are reported in parts per million (ppm) relative to the tetramethylsilane (TMS) peak (defined as 0.0 ppm). As the TMS standard was only present in the phenol solution, the NMR spectra in DMSO and acetic acid were shifted to align the solvent peaks with their tabulated values (acetic acid at 2.098 ppm and DMSO at 2.621 ppm). To facilitate a comparison of the ^1^H NMR spectra, they were scaled so that the best-resolved peak at 1.0 ppm, common to all spectra, had a consistent intensity. The ^1^H NMR spectra were recorded using an 11.7 T Bruker Avance 500 liquid-state spectrometer (Karlsruhe, Germany). Typical conditions were as follows: π/2 pulse, width of 10.5 μs, relaxation delay of 3 s, spectral width of 9 kHz, acquisition time of 1.7 s, and 800 scans. The integrated intensities were determined by spectrometer integration software (Bruker TopSpin 3.2) with an accuracy of ±1%. 

The solid-state ^13^C cross-polarization magic-angle spinning (CP/MAS) spectra were measured with an MAS frequency of 20 kHz and a relaxation delay of 10 s; 2000 to 10,000 spectra were accumulated. The contact time was set to 1 ms. The spectra were externally referenced to the signal of the carbonyl carbon of glycine. 

## 3. Results and Discussion

Initially, carbon solid-state NMR spectroscopy with magic-angle spinning (^13^C NMR MAS) was used to elucidate the chemical structure of the NPs. Figure 2 presents the spectra of the nylon-sputtered NPs and, for comparison, the pure nylon 6,6. Notably, the spectrum of the nylon 6,6 exhibits well-resolved, distinct peaks from 20 to 45 ppm. These peaks correspond to various CH_2_ groups, specifically αNH, αCO, βNH, γNH, and βCO, from left to right. A sharp C=O peak is evident at 172 ppm. Peak assignments were based on Ref. [46] and the chemical formula is shown in Figure 3.

Compared to the pure nylon 6,6, the NP spectra exhibit broad peaks, limiting the resolution to three distinguishable features, as follows: a faint C=N peak at approximately 170 ppm, a C=C peak, and a minor presence of C≡N within the 100 to 150 ppm range. Notably, a broad and robust peak dominates the CH_2_ region (0–100 ppm), likely containing additional contributions from C–O and C≡C. 

A comparison of the NP and pure nylon 6,6 spectra reveals significant changes in the NPs. Specifically, the double bonds (–C=O) in the original material have transformed into a network featuring –C–O– bonds within the NPs. Furthermore, carbon atoms within the NPs exhibit strong bonding, with a substantial portion (approximately a quarter of the CC bond intensity) forming double bonds (–C=C–). Some –C≡C– bonds are also present. Nitrogen atoms also display robust bonding within the NPs, including double and triple bonds.

Despite spinning the NP sample at 20 kHz and using five times more scans than for the original nylon, we could not resolve the MAS spectrum. This observation underscores the strong bonds of the NPs and the rigidity of their chemical groups. The broadness of the CH_2_ peak further corroborates the random, non-ordered structure of the NPs. The spectrum of the NPs was recorded with a much higher number of scans than the spectrum of the nylon in order to reduce noise. Nevertheless, the noise remained high due to the width of the peaks and the associated low intensity, which is further decreased by the low mobility of the chemical groups. 

These findings align with the previous measurements conducted using XPS and FTIR spectroscopy [15], which are summarised here:The elemental composition of the source nylon 6,6, composed of linear chains, obtained using XPS is 75.0% carbon, 12.5% oxygen, and 12.5% nitrogen. The NPs exhibited a comparable composition, although they underwent a slight denitrogenation. They comprise 81% carbon, 12% oxygen, and 7% nitrogen.Top of FormThe high-resolution XPS of the original nylon 6,6 showed the following bonds: primarily C–C, C–H with some C–N, and N–C = O. The C1s spectrum of the nylon-sputtered NPs was similar but slightly wider and showed the contribution of unsaturated carbon (C = C, C ≡ C) and C = N, C ≡ N, and C–O species, which are not present in the original polymer. The FTIR spectrum supported these findings, showing O-H and CH_3_ groups. The broadening of the peaks suggested a random structure neighbouring the groups corresponding to the observed peaks.

The solid-state NMR method described above yielded poor resolution in the spectra of the NPs due to the very limited mobility of the chains and chemical groups, suggesting a rigid network. Therefore, we investigated the soluble part of the NPs using high-resolution liquid-state NMR spectroscopy. This experimental approach exclusively captures data from the soluble fraction of the NPs, as the insoluble fraction, characterised by its enduring rigidity, produces peaks too broad for detection. The insoluble fraction was further studied by removing the solvent and examining the NPs with SEM.

To enhance the solubility and select the most suitable solvent for obtaining high-resolution NMR spectra, we dissolved the NPs in acetic acid, DMSO, and a 5 wt% phenol solution in water. Figure 4 compares the spectra of the NPs (bold lines) and the pure nylon 6,6 material. The solubility in the acetic acid and DMSO was comparable, while the NPs were less soluble in phenol than in either acetic acid or DMSO. The main reason for choosing acetic acid was because both the NP and the nylon spectra gave well-resolved peaks, which were easy to assign. Nevertheless, the distinctions between the NPs and nylon, as elaborated later, remain discernible in all solvents. Given that the nylon peaks in acetic acid (marked by stars in Figure 4) were the most sharply resolved, did not overlap with the solvent peaks, and could be completely assigned to corresponding chemical groups (indicated by the red curves), this solvent was used for further analysis.

To explore how the input power influences the structure of the NPs, we used three magnetron power levels, 40, 60, and 80 W; the corresponding NP samples are N40, N60, and N80, respectively. A comparison of the NMR spectra for these three NP variants dissolved in acetic acid is presented in Figure 5. The spectra exhibit a high degree of similarity. While there are subtle differences in intensity (at 3.7, 2.8, and 1.2 ppm) for N40, the fundamental conclusion is that varying the sputtering power per unit area of the target increases the deposition rate of the NPs, but does not significantly impact the structure of the soluble portion of the NPs. Given the labour-intensive process of generating a substantial quantity of NPs for experiments and the minimal alterations observed in the results, further investigations were conducted at the intermediate sputtering power of 60 W, focusing on the N60 NPs.

Figure 6 shows the liquid-state ^1^H NMR spectra of the N60 NPs and the pure nylon 6,6, which have been dissolved in acetic acid. The peak assignments use the Greek letters α, β, and γ to indicate the distance of the CH_2_ group from an NH or CO group. (For example, αNH indicates the CH_2_ group directly neighbouring the NH group and βCO is the second nearest CH_2_ to the CO group.) The assignments were made based on NMR nylon experiments published in Refs. [47,48] and are shown in Figure 3. In the spectrum of pure nylon 6,6, peaks corresponding to various chemical groups specified in the chemical structure of nylon 6,6 are discernible, as illustrated in Figure 6. The hydrogen bound to the N atom in the NH group (peak B) undergoes a chemical exchange process with the deuterated solvent and is unstable, broadening the corresponding NMR band. Therefore, the peak intensity is much lower than the NH abundance in the solution and this peak cannot be used in the NMR analysis. The liquid-state ^13^C NMR spectrum of the NPs (see Appendix A) is too weak and, therefore, cannot be used for evaluation. Therefore, no 2D NMR plots are possible.

The molar mass distribution of the soluble fraction of N60 was determined using GPC. The distribution displays three distinct peaks, as seen in Figure 7, all associated with relatively low molar masses in the context of polymers.

The relative intensity of these peaks, reflecting the proportion of molecules at the specified molecular weights, follows a pattern of 4:1:6 (from left to right in the main plot in Figure 7). The two narrow peaks on the left side of the main figure, with relative intensities of 4 and 1, and mean molar masses (Mn) of 155 and 294 g/mol (refer to Appendix A), indicate that half of the soluble molecules possess well-defined molecular weights. In contrast, the position and shape of the peak on the right side (from 500 to 1000 g/mol) correspond to oligomeric molecules. Notably, the slight undulation observed at the lower molecular weight side of the first peak represents the contribution from low molecular weight impurities, such as residues of the monomers or the solvents used in the GPC process.

The liquid- and solid-state NMR experiments show that the NPs consist of a densely cross-linked insoluble fraction and a soluble fraction. Using SEM, the sizes of the as-deposited NPs, and those after dissolution in the tested solvents and subsequent drying, were measured (an example is shown in Figure 8). Images of the NPs prepared at different powers and dissolved in different solvents are presented in Appendix A. The sizes of N40, N60, and N80 in acetic acid and of N60 in different solvents did not differ significantly from the as-deposited state and were around 44 ± 2 nm (Appendix A). The NPs have a cauliflower structure and the size has a relative error of about 5%, comparable to the difference in the values. 

The N60 NPs were also tested for thermal stability. Samples were tempered for 20 min at 100, 200, 300, 400, and 500 °C (a different piece for each temperature). Although the NPs emitted an odour during heating, indicating the evaporation of some low molecular weight fragments, the NP sizes did not differ from that of the non-heated sample (see Appendix A). From these results, it may be concluded that the insoluble part creates a rigid, strongly bound (network) skeleton that is inert to dissolution and heating. The soluble part is incorporated into this skeleton and may be washed out by a solution or released using heat. 

Two components were detected in the NPs. While the primary insoluble fraction appears to be composed of disordered random fragments of the original nylon molecule tightly bound together, the soluble fraction seems to have a different structure. Thus, we propose the chemical structure for the dominant portion of the soluble part of the NPs. The ^1^H NMR spectra indicate the presence of the chemical groups listed in Table 1. No oxygen was found in the soluble fraction. 

Table 1 presents the chemical groups corresponding to the ^1^H NMR spectrum in Figure 6 and reports their approximate ratios. These ratios were determined by integrating the peaks and dividing the result by the number of hydrogens in the respective chemical group. However, the precision of these calculations is limited. The peaks, and consequently their integral intensities, are relatively small. Moreover, some chemical groups share the same position in the spectrum, there is overlap with the solvent peak, and the intensity of the NH and NH_2_ groups is underestimated due to exchangeable hydrogen. Despite these challenges, it is possible to form a conceptual representation of the structure of the soluble portion of the NPs. This fraction is primarily composed of CH_2_ chains, predominantly terminated with CH_3_ groups and, to a lesser extent, with NH_2_ groups. The CH_2_:CH_3_ ratio is in the range of 3–5 CH_2_ units for every CH_3_ group. Additionally, some NH groups are incorporated within the CH_2_–CH_2_ chains.

Based on the GPC experiments, 40% of the soluble NP molecules comprise ten carbon or nitrogen units, while the remaining 60% are larger molecules of 25 to 95 carbon or nitrogen units. To maintain the previously calculated CH_2_:(end of the chain) ratio, these molecules must exhibit a highly branched structure, which is expected in NPs produced through sputtering. Identifying the specific branching element is challenging, particularly in ^1^H NMR spectra, where only hydrogen signals are detected. The ^13^C NMR spectra cannot be used because their peaks are too weak to be properly evaluated. Direct signals from quaternary carbons are absent in the ^1^H NMR spectra, but their presence can be inferred indirectly through neighbouring signals. However, no such signals are observed in the spectrum, suggesting the absence of quaternary carbons in the sample. Similarly, tertiary nitrogen cannot be directly detected, but the spectrum does not exclude its presence. Because there is no signal in the NMR spectrum from quaternary C, and CH is only found in the neighbourhood of the end group NH_2_, the branching likely involves triple-bonded nitrogen. In summary, the soluble part of the sputtered NPs consists of a network of chains made of CH_2_ and some NH groups, branching with nitrogen. The chains are terminated with CH_3_ or NH_2_. A schematic of the structure of the soluble fraction of the NPs is shown in Figure 9.

SEM images, exemplified by Figure 8 (full set is shown in Appendix A), provide clear evidence that the soluble molecules adhere to or reside within the NPs, with no observable continuous soluble phase, as seen in other systems [49]. Furthermore, this soluble fraction does not influence the size of the NPs. The size of the sputtered NPs remains unchanged even after the dissolution of the soluble component or heating of the NPs, accompanied by the release of low molecular weight molecules. This observation likely results from the considerably smaller quantity of soluble material within the NPs compared to the insoluble fraction. This notion is supported by the observation that, during the NMR experiments, although a substantial amount of insoluble sediment settled at the bottom of the cuvettes, the NMR signal from the dissolved monomers and oligomers remained low. Consequently, the soluble fraction is significantly smaller than the insoluble fraction.

## 4. Conclusions

The structure of nylon-sputtered NPs has been studied using liquid- and solid-state NMR. While these NPs were previously examined in Ref. [15], the size of the molecules had not been determined. Using NMR and GPC, the composition of the NPs was investigated in terms of a soluble, low molecular weight fraction and an insoluble, gelated fraction. This research aimed to explain the reflection of plasma polymer NPs from the substrate upon high-speed impact, a behaviour not observed in metal NPs, suggesting a rubber-like character. It was confirmed that each NP is composed of a single fully-gelated structure infused with a small amount of low molecular weight material disconnected from the main network, which may be released using solvent extraction or heat, without having a measurable effect on the size of the NP.

The NPs were produced using a semi-hollow magnetron GAS. The plasma fragmented the nylon 6,6 target, and highly cross-linked NPs were created. This process transformed about a quarter of the C-C bonds into C=C bonds. Additionally, C≡C, C=N, and C≡N bonds form. Furthermore, most of the C=O double bonds converted to network-making single bonds that further contribute to the crosslinking of the plasma polymer. The soluble part can be dissolved in acetic acid, DMSO, and phenol and is, therefore, detectable by high-resolution liquid-state NMR spectroscopy. It consists of highly branched CH_2_ chains, terminated with CH_3_ or NH_2_ groups and branched with the help of nitrogen. The NPs prepared at different sputtering powers did not differ significantly in structure; only their quantity increased with applied power.

## Figures and Tables

**Figure 1 nanomaterials-14-00497-f001:**
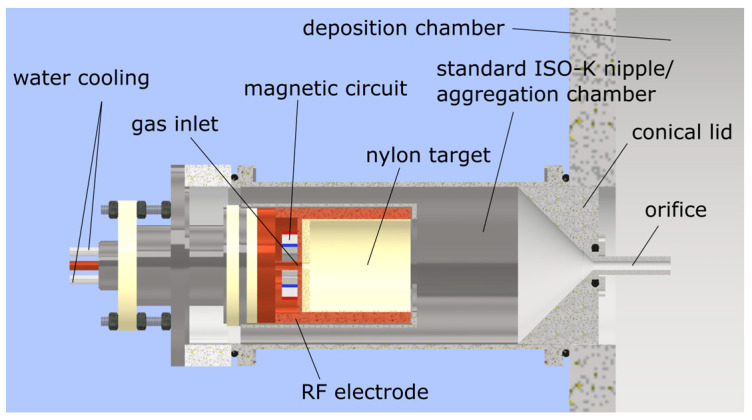
Schematic of the gas aggregation source (GAS) used to produce nylon-sputtered plasma polymer nanoparticles (NPs).

**Figure 2 nanomaterials-14-00497-f002:**
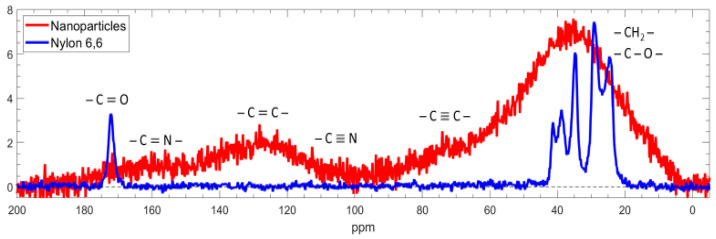
^13^C MAS spectra of nylon-sputtered N60 particles (particles sputtered at 60 W) and original nylon 6,6. The sharp peaks between 20 and 40 ppm are the CH_2_ groups of the nylon 6,6 described in the text. The intensity of both spectra was normalised to the highest intensity peak.

**Figure 3 nanomaterials-14-00497-f003:**
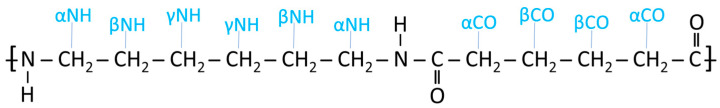
Chemical formula of nylon 6,6 with assignments of the nylon groups.

**Figure 4 nanomaterials-14-00497-f004:**
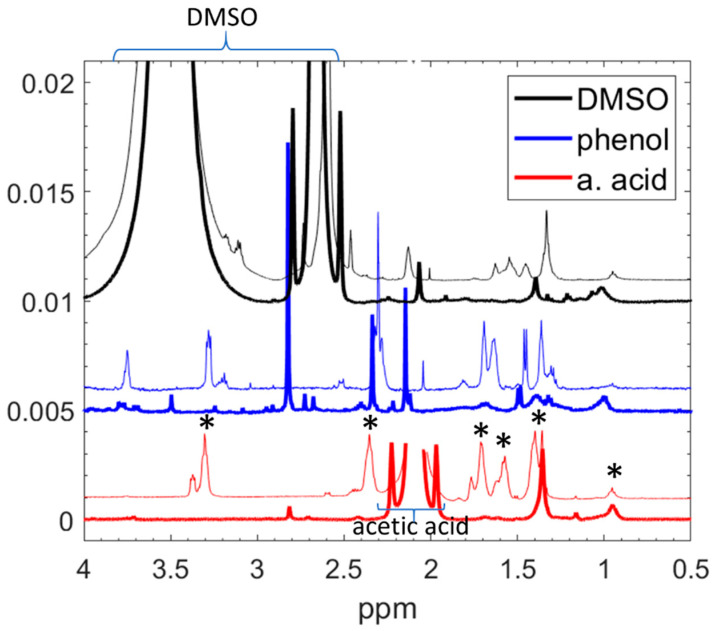
Comparison of the liquid-state ^1^H NMR spectra of NPs prepared at 60 W (bold lines) and the original nylon 6,6 (thin lines) dissolved in different solvents. For comparison, the intensities are normalised to the CH_3_ peak (0.95 ppm). The peak of acetic acid (at 2.1 ppm) is truncated for better transparency.

**Figure 5 nanomaterials-14-00497-f005:**
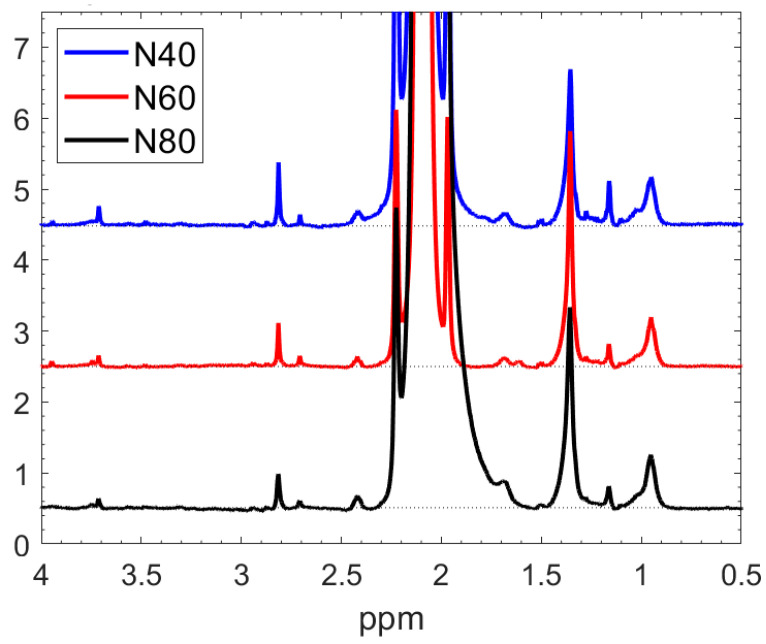
Comparison of the liquid phase NMR spectrum of the NPs (prepared at different sputtering powers) in acetic acid. The dot lines are the baselines Assignments of the peaks will follow in Figure 6.

**Figure 6 nanomaterials-14-00497-f006:**
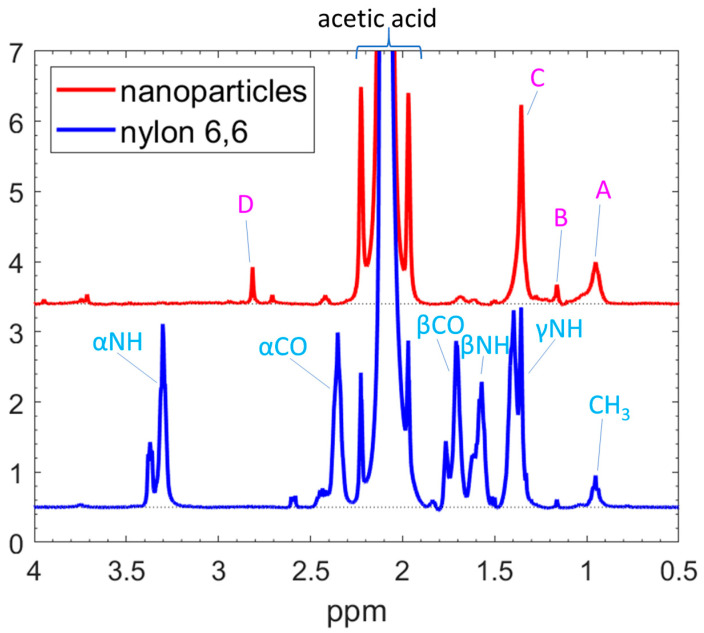
^1^H liquid-state NMR spectra of the nylon 6,6 and N60 NPs in acetic acid. Only the aliphatic region is shown; the whole spectrum and the spectrum of pure acetic acid are given in Appendix A. The intensities are normalised to the γNH peak, and the x-axis is calibrated with the peak of acetic acid at 2.098 ppm, as is described in the experimental section. The nylon 6,6 peaks (blue text) are labelled according to the chemical formula in Figure 3 and assigned according to Refs. [47,48]. The NP peak (A–D) assignments are explained in Table 1. The dot lines are the baselines. The peaks split due to the different tacticity of the molecules.

**Figure 7 nanomaterials-14-00497-f007:**
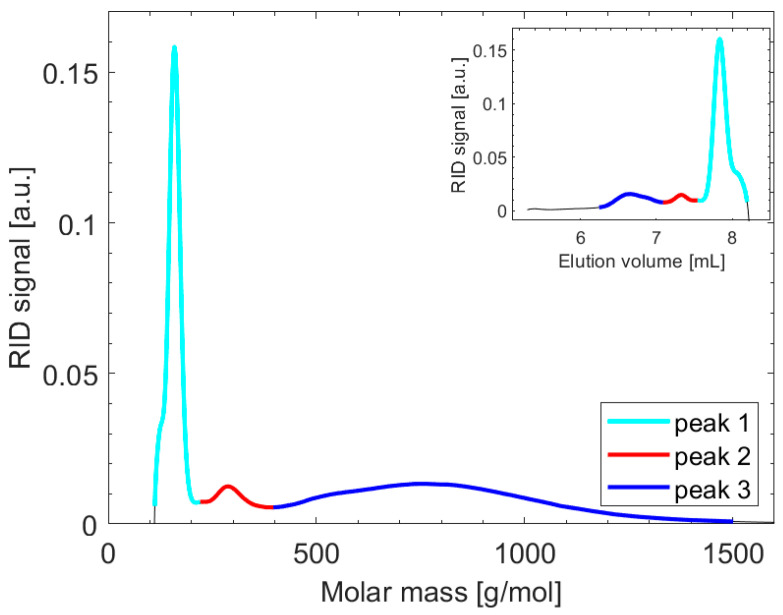
Molar mass distribution of the soluble fraction of the N60 NPs after filtration was determined from the GPC curve (inset) based on the polystyrene standard.

**Figure 8 nanomaterials-14-00497-f008:**
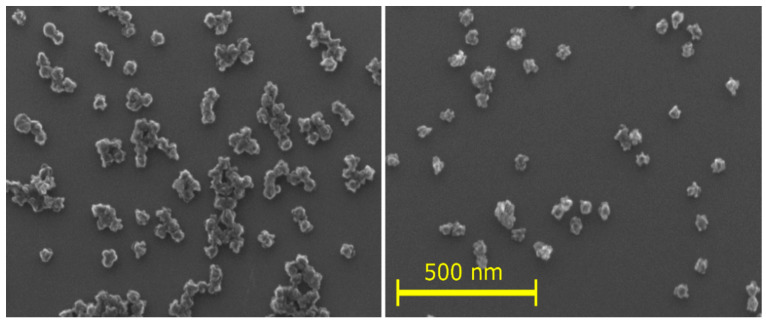
Example of SEM photography for N60 dissolved in acetic acid (**left**) and as-deposited N60 (**right**).

**Figure 9 nanomaterials-14-00497-f009:**
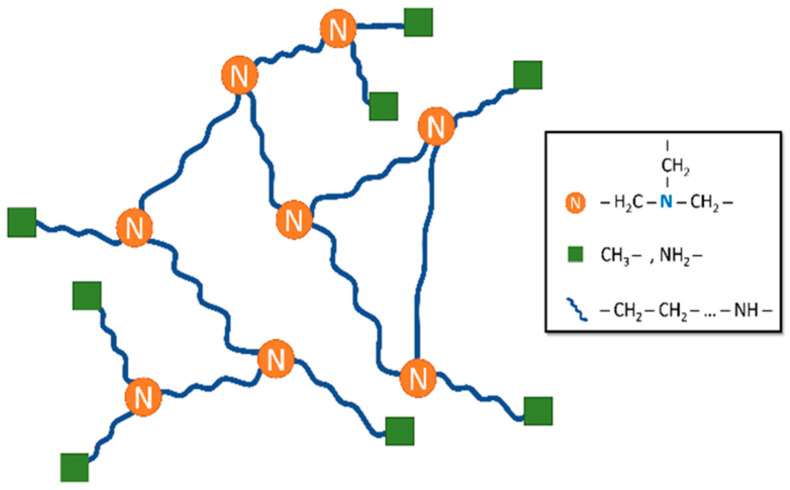
Suggested schematic of the chemical structure of the soluble fraction of the nylon-sputtered NPs.

**Table 1 nanomaterials-14-00497-t001:** NMR peaks of the soluble fraction of NPs. The NMR signals were assigned based on Ref. [47]. The relative molar abundance of the chemical groups within the soluble fraction of the NPs, when dissolved in acetic acid, was determined based on the intensity of the corresponding NMR peaks. In cases where a single peak corresponds to two chemical groups, two approximate ratios were calculated depending on the number of hydrogens present in each group. Due to the low intensities, the determined ratios are only approximate.

Assignment in Figure 6	Position in the Spectra (ppm)	Chemical Group	Relative Molar Abundance of the Chemical Groups
A	0.96	**CH**_3_–C–	5
B	1.2	**NH**_2_–C–, –C–**NH**–C–	>1–2
C	1.3	–C–**CH**_2_–C–	20
acetic acid	2.1	**CH**_3_–N–, (acetic acid)	approx. 2
D	2.8	–N–**CH**_2_–C–, –C–**CH**–NH_2_ | CH_3_	2.5–5

## Data Availability

Data are available on request from the authors.

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
