# Peer review of "Insoluble Network Skeleton and Soluble Components of Nylon 6,6-Sputtered Nanoparticles: Insights from Liquid-State and Solid-State NMR Analysis"

_nanomaterials, 2024, doi:10.3390/nano14060497_

Round 1
Reviewer 1 Report
Comments and Suggestions for Authors
Please add the words/contents in lines 47, 53, and 223, do not only put references.
Please revise introduction section, especially last phrase without references and the purpose. Readers may not understand why to study it.
Authors may try high resolution 13C MAS spectrum of sputtered-nylon N60 particles by increasing scan times.
Please explain the mechanism of dissolving in acetic acid? I do not think that acetic acid is better solvent for nylon N60 particles, may be DMSO or phenol?
Please revise English writings.
Comments on the Quality of English Language
Authors need correct some sentences and polish English writing.
Author Response
Dear reviewer,
first of all, we would like to thank you for your interesting comments and suggestions related to our manuscript. We have modified our manuscript taking into account these comments. The detailed replies can be found in the attached pdf and all modifications are highlighted in the revised manuscript. We hope that the changes that we have made clarified all the points raised by you.
On behalf of co-authors,
Julie ŠÅ¥astná

Reviewer 2 Report
Comments and Suggestions for Authors
The abstract provided contains valuable information but could benefit from some improvements for clarity and coherence. This revision aims to streamline the language, provide clearer descriptions of the methods and findings, and improve the overall flow of information.
The investigation confirmed the presence of suspected tight binding and networking among the nanoparticles, elucidating a soluble monomer/oligomer phase alongside. The chemical structure of this low-molecular-weight phase was proposed, contributing to a better understanding of the material. Notably, the observed structure lends support to the hypothesis of the rubber-like characteristics of plasma polymer nanoparticles, which leads to their reflection from the substrate upon high-speed impact.
Line 72: “Top of Form”?
Following lines (73 – 79) seem to be detailed data analysis, that should not belong to Introduction section, but to Results.
Lines 84-84: “1 mm in the cylinder.” What do the authors mean?
Some abbreviations for units should be described, such as sccm for flow rate, as these are not SI units. It is advisable using SI units whenever possible. See for instance afterwards, flow rate expressed as mL/min (line 122), which is the type of unit the readers expect.
Is the 13C MAS spectrum (Fig. 1) the best recorded? The noise is too high to claim much from the humps present.
Lines 228-239: The N-H can actually be integrated properly if the authors would run the 1H NMR with a longer relaxation time, around 2-4 s.
Regarding remark in Fig 5 – the peaks should be checked for shape with a better shimming of the instrument; see the satellites for the solvent peak, they are not symmetrical.
Figure 6 must be re-evaluated; is the structure known, as presented? Have any NMR spectra been ever reported? Did the authors check if those shifts match? Are there any other shifts above 4 ppm?
The authors are encouraged to retake the NMR spectra and run extensive analysis to corroborate with Fig 6. 1H NMR, 13 C NMR and at least some 2d (HSQC for instance) should make identification easier.
Elemental analysis is also necessary. This would point out if the structure proposed is viable or not.
Comments on the Quality of English Language
A few errors should be removed during proofreading stage.
Author Response

(The authors gave the same response as above.)

Round 2
Reviewer 1 Report
Comments and Suggestions for Authors
Lines 45, 55, and 260, in what? based on what? Please add the contents with the reference.
Comments on the Quality of English Language
Extensive editing of English language is required. It is unacceptable that authors had used ChatGPT for English writing correction.
Lines 45, 55, and 260, in what? based on what? Please add the contents with the reference.
Reviewer 2 Report
Comments and Suggestions for Authors
Some of the issues raised have been addressed. However, some observations point out to essential issues within the draft. In short, while 13C NMR could be weak, having no peaks after multiple runs, corroborated with almost no result from 1H NMR spectrum, implies that the phrase "soluble part of" needs to be double-checked. To me, this points out to insoluble materials and hence, no conclusions can be drawn from the spectra presented (other than that they belong to the residual solvent peaks). Sadly, I must insist on other analysis techniques in order to claim the structures proposed by the authors.
Additionally, in Fig. 4, the legend covers DMSO spectrum and should be placed in a neutral region to not cover any significant peaks.
Figure 9 should also depict some clear proposed structures, the most likely to be found.
I also suggest that the original fids be included in the SI or the evaluation materials for reviewers to attest the assignments.
Comments on the Quality of English Language
No major flaws detected, but a careful proofreading of the manuscript is needed.
